# Acquisition and Processing of UAV Fault Data Based on Time Line Modeling Method

**Tao Yang [1], Yu Lu [2,\*], Hongli Deng [1], Jiangchuan Chen [2] and Xiaomei Tang [3]**

[1] Education and Information Technology Center, China West Normal University, Nanchong 637001, China
[2] School of Computer Science, China West Normal University, Nanchong 637001, China
[3] Nanchong Institute of Educational Sciences, Nanchong 637000, China
\* Correspondence: luyu@stu.cwnu.edu.cn

**Abstract:** The number of Unmanned Aerial Vehicles (UAVs) used in various industries has increased exponentially, and abnormal detection of UAVs is one of the primary technical means to ensure that UAVs can work normally. Currently, most anomaly detection models are trained using on-board logs from drones. However, in some cases, using these logs can be problematic due to data encryption, inconsistent descriptions of characteristics, and imbalanced positive and negative samples. Consequently, the on-board logs of UAVs may not be directly usable for training anomaly detection models. Given the above problems, this paper proposes a Time Line Modeling (TLM) method based on the UAV software-in-the-loop (SITL) simulation environment to obtain and process the on-board failure logs of drones. The Time Line Modeling method includes two stages: the Fault Time Point Anchoring Method and Fault Time Window Stretching Method. First, based on the SITL simulation environment, multiple flight missions were constructed. Failures of several common components of UAVs are designed. Secondly, the fault's initial location and end location are determined by the method of Fault Time Point Anchoring, and the original collection of tagged UAV's on-board data is realized. Then, in terms of data processing, the features that are not universal are removed, and the flight data of the UAV is optimized by using the data balance method of Time Window Stretching to achieve the balance of normal data and abnormal data. Finally, use of algorithms such as Sequential Minimal Optimization (SMO), Random Forest (RF), and Convolutional Neural Network (CNN) were used to experiment with the processed data. The experimental results showed that the data set obtained based on this method can be effectively applied to the training of machine learning-based anomaly detection models.

**Keywords:** UAV; UAV anomaly detection; data balance; SITL

## 1. Introduction

With the rapid development of control theory, aerodynamic technology, navigation technology, and communication technology, the application of drones in agricultural image acquisition [1], traffic monitoring [2], environment detection [3], etc., is gradually increasing. According to statistics from the Civil Aviation Administration of China, at the end of 2021, there were 830,000 drones registered with their real names in China alone [4]. However, due to the drone's structural complexity and the flight environment's variability and instability, the flight of the drone is often affected by external factors such as weather and buildings, resulting in potential safety hazards. After the drone fails, if no remedial measures are taken, it may cause the drone to crash and cause loss of life or property to nearby people. For example, the failure of GPS will lead to the crashing of drones. According to the report from the British Aviation Accidents Committee, a British survey drone fell on a roof due to GPS failure [5]. The current security issues of UAVs have attracted widespread attention in the academic community, and the anomaly detection of UAVs has become one of the current research topics. Figure 1 shows a typical UAV anomaly detection network architecture. The

fault detection model can be deployed on the GCS, and when the fault detection model receives the log, it will be inputted to the fault detection model for judgment in order to obtain the current state of the drone.

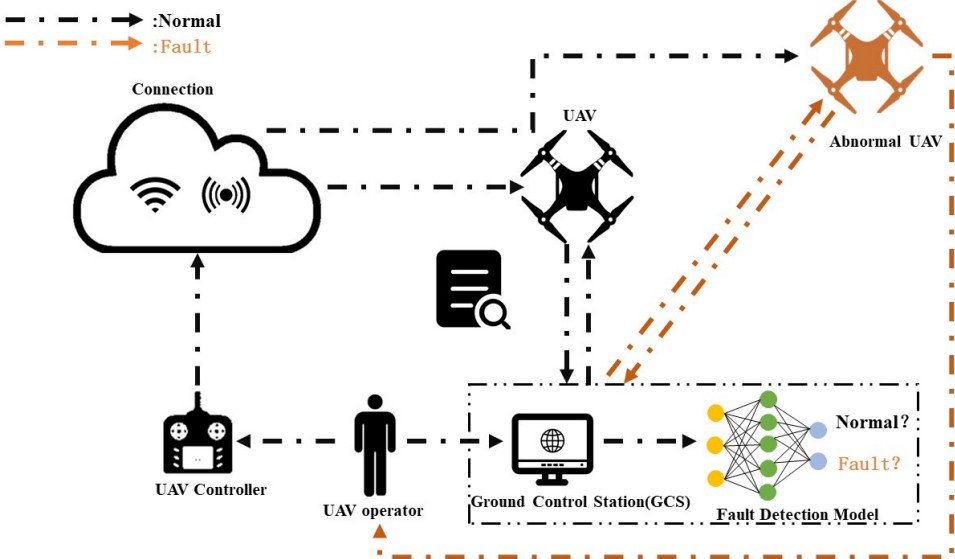

**Figure 1.** A typical drone anomaly detection network architecture.

With the development of artificial intelligence technology, machine learning methods are widely applied to the fault detection of UAVs. Machine learning models based on drone flight logs require a large number of samples for training, but due to the following reasons, it is often challenging to obtain drone flight data that can be directly used for model training:

- The logs of commercial drones are not open source and cannot be obtained. The flight log of the open-source UAV includes information such as the flight status, telemetry log, sensor information, communication information, and hardware status of the UAV within a time window T0~T1. Its data volume is large, the dimension of features is high, and the description of features is not uniform; that is, heterogeneous data from different sensors and features are difficult to process.
- During the flight of the UAV, the time of abnormal flight is much lower than that of normal flight, which makes the number of samples when the UAV fails much lower than that of normal flight, and it is challenging to obtain abnormal data. Due to the extreme imbalance of the data, it is difficult for models to learn the abnormal data, so the unbalanced flight log cannot be directly used for the fault detection of the UAV.

Given the above problems, this paper proposes a method based on Time Line Modeling to acquire and process UAV fault flight data in a simulation environment. This paper constructs the flight mission of the UAV in the simulation environment and realizes the original collection of the flight log. The time-related features and non-universal features are removed, and the feature selection of UAV data is realized. The initial and end positions of the fault data are determined using the Fault Time Point Anchoring Method. The Time Window Stretching method is used to effectively supplement the abnormal data so that the data becomes balanced. Machine learning and deep learning algorithms, such as K-NN, AdaBoost, random forest, Naive Bayesian, Decision Tree, CNN, etc., are used to evaluate the obtained data sets. The experiment proved that the acquisition method of UAV fault data in the simulation environment proposed in this paper is effective in UAV fault detection.

In the rest of this paper, Section 2 introduces the related work. Section 3 introduces the construction method of UAV flight tasks in the simulation environment. Section 4 introduces the Time Point Anchoring method and the balance processing method of positive

and negative samples. Section 5 evaluates the processed data using machine learning algorithms and convolutional neural networks. Section 6 concludes the work.

## 2. Related Work

This section reviews the research related to the abnormal detection of UAVs. Regarding anomaly detection models, it reviews related works on machine learning-based and deep learning-based models. In terms of data set acquisition, the real and simulated UAV flight data acquisition methods are reviewed.

### 2.1. Related Fault Detection or Intrusion Detection Models

#### 2.1.1. Machine Learning-Based Models

In 2017, Baskaya et al. [6] proposed using the Support Vector Machine (SVM) algorithm to detect engine faults, the process of training and testing using gyroscope and accelerometer data, and the principal component analysis algorithm as a method to reduce the spatial feature dimension. The authors used a MAKO drone model to generate data to test the designed algorithm. In 2019, Benini et al. [7] used Linear Discriminant Analysis (LDA) as a model and acceleration information from an Inertial Measurement Unit (IMU) as data to detect actuator failures in drones. In 2022, Cabahug et al. [8] implemented the fault detection of drones using the K-Means algorithm and the vibration data of drones. The authors used custom hardware to collect vibration data from UAV propellers. The K-Means algorithm was used to cluster the three states of the drone.

#### 2.1.2. Deep Learning-Based Models

In 2021, Farrukh et al. [9] used the bidirectional long short-term memory neural network (BiLSTM) to match the attack according to the existing attack of the model: if the match is successful, a warning is issued; if no attack is matched, the data is sent to an anomaly detector based on the Local Outlier Factor (LOF). If the anomaly detector detects an anomaly, it will issue an alarm and collect the data of this attack instance, then mix the new instance data with the old data and send them to the BiLSTM for training to complete incremental learning. The authors used the CSE-CIC-IDS2018 data set [10] to test the model. In 2022, Al-Haija et al. [11] used a deep convolutional neural network to detect malicious threats from drones in Wi-Fi traffic. Two-way and one-way drone traffic data were used for model training and testing. In 2022, Tlili et al. [12] proposed to use variants of the Long Short-Term Memory neural network (LSTM) to detect anomalies and faults simultaneously. This model consists of the main branch and two sub-branches. The main branch produces the general LSTM encoder architecture. The two sub-branches are used to detect anomalies and faults.

### 2.2. Related Data Acquisition Methods

#### 2.2.1. Real UAV Data Acquisition Method

In 2020, Zhao et al. [13] proposed that consumer drones are usually used in civilian environments, and traditional physics-based methods will become ineffective in some cases. They propose to conduct drone intrusion detection through encrypted Wi-Fi data traffic. In 2021, Keipour et al. [14] based on the Carbon Z T-28 fixed-wing UAV designed four kinds of faults: disabling the engine, disabling the elevator, disabling the rudder, and disabling the aileron, realizing the collection of UAV flight logs in the fault state. In 2022, Ahmed et al. [15] conducted a vulnerability analysis of drones used in the education sector. The drone platform used was Ryze Tello TLW004. This data set contains Wi-Fi Deauthentication Attack, WPA2-PSK Wi-Fi cracking attack, and a Tello API vulnerability attack.

#### 2.2.2. Simulated UAV Data Acquisition Method

In 2015, Moustafa et al. [16] developed the UNSW-NB15 data set, used three virtual servers as the experimental environment, and captured 49 different network traffic characteristics. It included nine different types of network attacks and some normal data flow

characteristics. In 2021, Whelan et al. [17] proposed that GPS spoofing and jamming are the most common attacks against drones, but conducting these experimental studies in many fields may take much work. The development of this data set used the Holybro-S500 drone platform and Hack RF to jam and deceive the drone. They implemented log collection in the case of UAV jamming and spoofing. In 2021, Ahmed et al. [18] created an IoT test platform for the medical industry and developed an IoT intrusion detection data set called ECU-IoHT. This data set reflects different types of network attacks, which contains the characteristics of different networks under various attacks.

It can be seen that when studying UAV fault detection or anomaly detection models, the research on the model is subject to the data. However, the UAV data has problems, such as difficulty accessing drone logs due to data encryption, inconsistent descriptions of drone logs, and unbalanced positive and negative samples. These problems limit the further development of anomaly detection models. This paper proposes a method for acquiring and processing UAV fault data in a simulation environment, that is, a method based on Time Line Modeling (TLM). Experiments proved that the UAV fault data obtained by the TLM method can be effectively applied to the training of machine learning models.

## 3. The Design of Simulation Environments and Simulation Tasks

The workflow of the TLM method is illustrated in Figure 2. First, the UAV simulation task is constructed, followed by simulating common faults that could occur in UAVs. Finally, the Fault Time Point Anchoring Method and the Fault Time Window Stretching Method are used to mark and balance the obtained UAV data, respectively. The specific implementation of the TLM method is introduced in the following Sections 3 and 4.

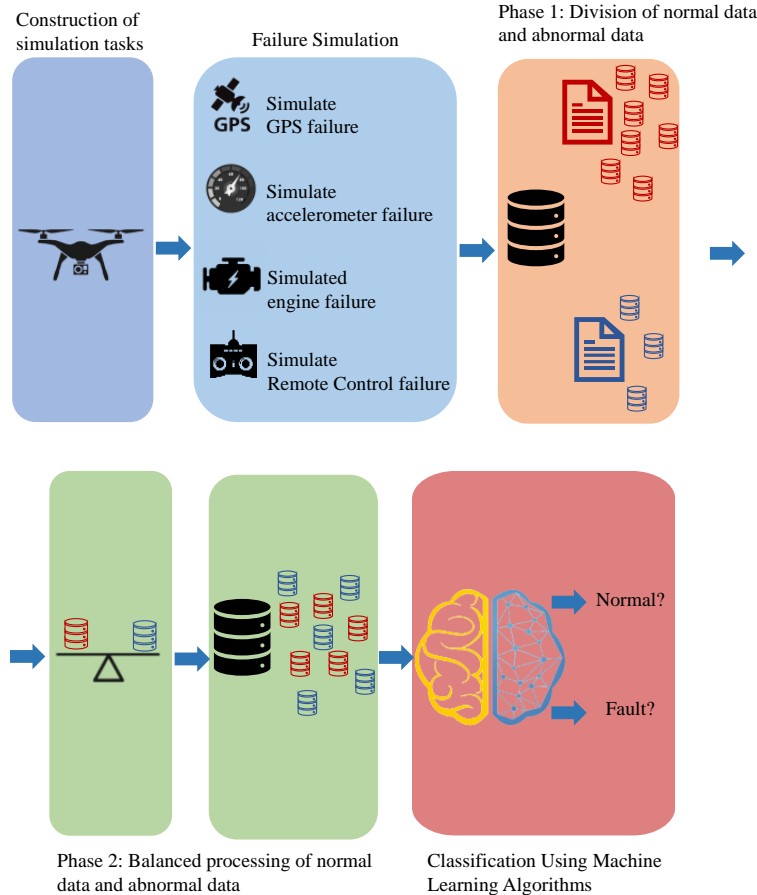

**Figure 2.** Flowchart of the TLM method.

*3.1. Simulation Environment*

(1)　Simulation platform

This paper uses Software-In-The-Loop (SITL) in ArduPilot to simulate the flight of the UAV. ArduPilot is a versatile open-source autopilot system that can provide autopilot services for multi-rotor UAVs, fixed-wing UAVs, rovers, ships, etc.

(2)　Ground Station

In this paper, the QGroundControl ground station is used to visualize the real-time position of the UAV. This paper uses the Python interface of Dronekit to realize the drone's control, including take-off, going to the next waypoint, outputting the current flight status of the drone on the console, returning, landing, and other functions.

(3)　Communication Control

Dronekit-Python is an open-source Python library used to control the flight of drones. It uses the MAVLink protocol to communicate with drones through serial ports. The software-in-the-loop simulation of the drone runs on an Ubuntu 18.04 system based on VMware Workstation 16.2.3, and the ground station runs on a Windows 11 laptop with 16 GB RAM, 11th Gen Intel(R) Core (TM) i7-11800H @2.30 GHz.

*3.2. Construction of UAV Simulation Flight*

In this section, the construction of the UAV flight is introduced. This paper identifies four simple flight paths. The four flight trajectories are shown in Figure 3. A linear interpolation method is used to insert more waypoints between two waypoints. The interpolation algorithm is shown in Algorithm 1.

---

**Algorithm 1:** Linear interpolation algorithm between two waypoints points

---

Input: raw latitude and longitude points;
Output: longitude and latitude points after linear interpolation;
for i in len(raw data points):
　　Calculate the straight line ($LineLon_i$) passing through two waypoints;
for j in range(PointNum):
X_New= Equidistant points at highest and lowest latitudes;
　　　　Y_New=Bring X_New into the equation of $LineLon_i$;
　　　　Save X_New, Y_New to txt file;
End.

---

$$LineLon_i = k * LineLat_i + b \tag{1}$$

Among them, $k$ refers to the slope of two points $(Lat_i, Lon_i)$, $(Lat_{i+1}, Lon_{i+1})$. $b$ is the coefficient. $Lat_i, Lon_i$ refers to the currently processed longitude and latitude, and $LineLon_i$, $LineLat_i$ refer to the newly generated longitude and latitude. For example, the straight line passing through points $A(-35.3632621, 149.1652374)$ and $B(-35.36328381, 149.16306103)$ is $LineLon_i = 100.24735145735254 * LineLat_i + 3694.238601803339$.

The matrix of Equation (2) shows the coordinate points generated by the interpolation algorithm.

$$\begin{bmatrix} \boldsymbol{LineLat_i} & \boldsymbol{LineLon_i} \end{bmatrix} = \begin{bmatrix} -35.3632621 & 149.16523739 \\ -35.36326365 & 149.16508193 \\ -35.3632652 & 149.16492648 \\ \vdots & \vdots \\ -35.36328226 & -35.36328381 \\ -35.36328381 & 149.16306102 \end{bmatrix} \tag{2}$$

In order to simulate the altitude change of the UAV, this paper generates altitude in a random range and uses the Savitzky-Golay filter algorithm [19] to smooth them. The

comparison effect before and after smoothing is shown in Figure 4; the blue line refers to the randomly generated height within the range of 15 to 25 m. It can be seen that its change range is huge. The red line refers to the altitude after smoothing. It can be seen from the figure that the altitude change of the UAV smoothed by the algorithm is more gentle, which can simulate the height change of the actual UAV.

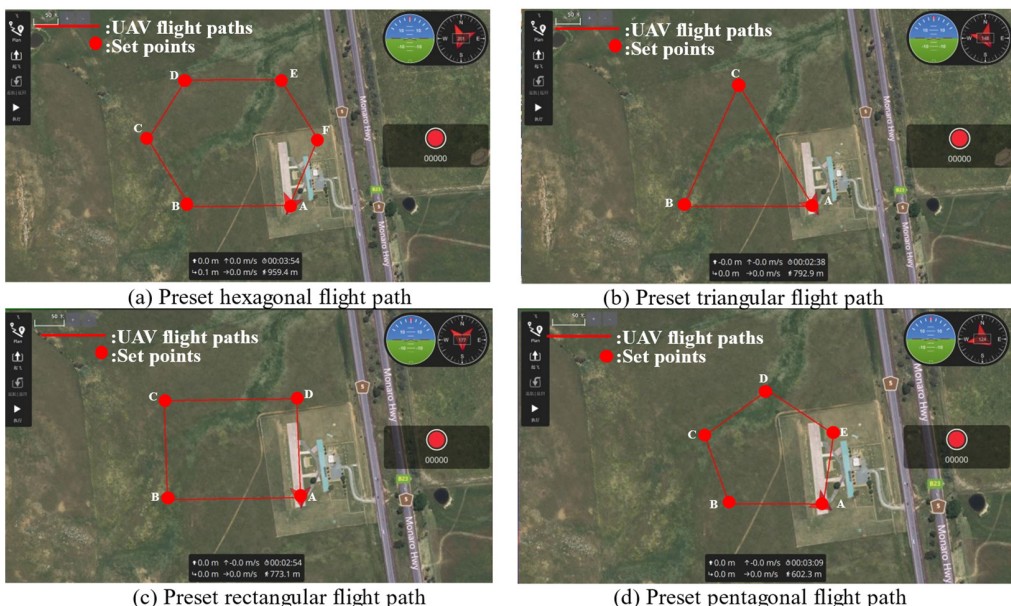

(a) Preset hexagonal flight path

(b) Preset triangular flight path

(c) Preset rectangular flight path

(d) Preset pentagonal flight path

**Figure 3.** UAV 2D flight trajectory map.

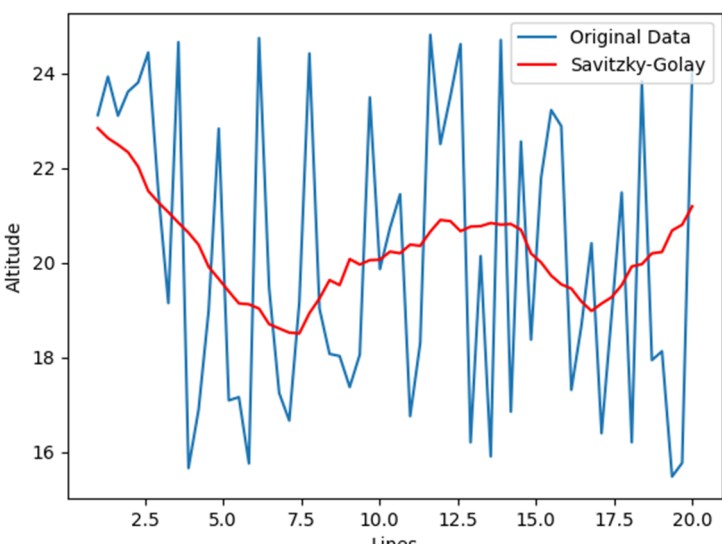

**Figure 4.** Altitude comparison of waypoints before and after using the Savitzky-Golay algorithm.

This article uses a 3-dimensional scatter diagram to display the changes in the waypoints before and after using the linear interpolation algorithm and Savitzky-Golay filter algorithm. As shown in Figure 5, the blue line is the change of longitude, latitude, and altitude, and the orange line is the change of longitude and latitude. Figure 5a shows that there are only six waypoints before the waypoint is inserted, and the altitude changes drastically, which cannot genuinely simulate the actual flight trajectory of the UAV. As shown in Figure 5b, after using the interpolation algorithm and the Savitzky-Golay algorithm, there are 90 waypoints in the figure, and the change of altitude tends to be gentle.

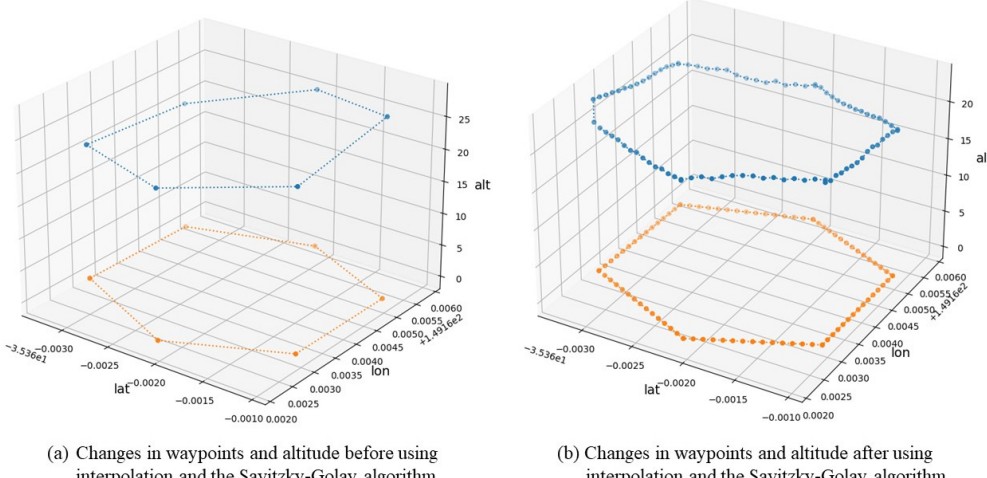

(a) Changes in waypoints and altitude before using interpolation and the Savitzky-Golay algorithm

(b) Changes in waypoints and altitude after using interpolation and the Savitzky-Golay algorithm

**Figure 5.** Changes in waypoints before and after using interpolation and the Savitzky-Golay algorithm.

This paper uses Dronekit to control the UAV for four autonomous flights. The simulations of GPS failure, Accelerometer failure, Engine failure, and Remote-Control System failure were performed simultaneously. The flight path of the UAV under QGroundControl is shown in Figure 6. Each point in the figure means: GPS failure occurs from point A to B, and GPS returns to normal from point B to C; the accelerometer failure occurs from point C to D, and the accelerometer returns to normal from point D to E; the engine failure occurred from point E to F, and the engine returned to normal from point F to G; from point G to H, a Remote-Control System failure occurred. For example, in Figure 6a, a GPS failure occurs from point A ($-35.3615, 149.1625, 21.9135$) to point B ($-35.3610, 149.1644, 21.2618$). Accelerometer failure occurred from point C ($-35.3611, 149.1650, 19.9449$) to D ($-35.3617, 149.1655, 18.7668$). Engine failure occurred from point E ($-35.3622, 149.1657, 18.6322$) to F ($-35.3629, 149.1654, 19.6512$). From point G ($-35.3631, 149.1652, 19.8433$) to H ($-35.3632, 149.1638, 21.7933$) a remote control failure occurred.

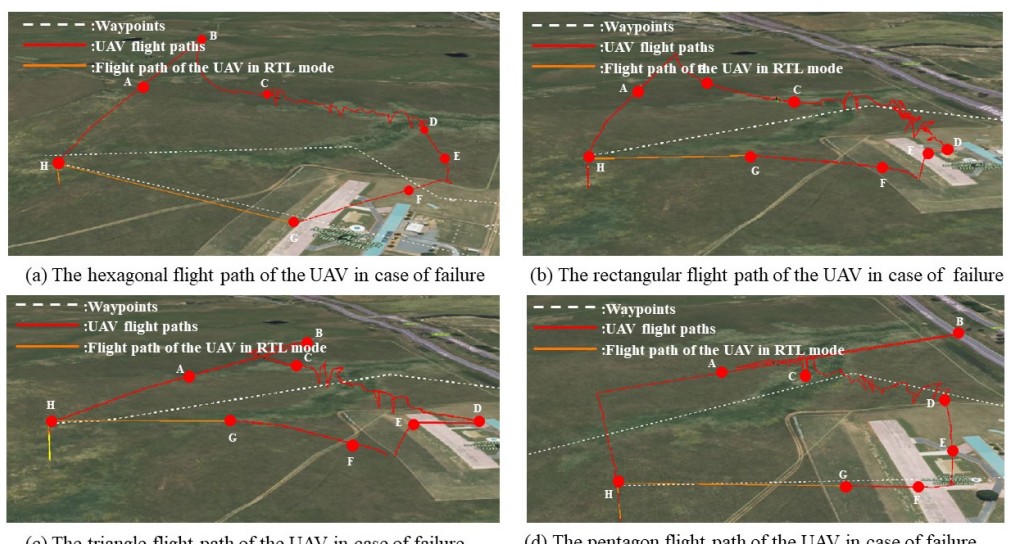

(a) The hexagonal flight path of the UAV in case of failure

(b) The rectangular flight path of the UAV in case of failure

(c) The triangle flight path of the UAV in case of failure

(d) The pentagon flight path of the UAV in case of failure

**Figure 6.** The trajectory of the UAV under QGroundControl during fault simulation.

### 3.3. GPS Parameter Adjustment

The Number of Visible Satellites is an important indicator of whether the UAV experienced GPS failure. In the simulation experiment, the "Number of Visible Satellites" variation range is too obvious. When the GPS signal is normal, the value of "Number of Visible Satellites" is always 10; when the GPS signal is abnormal, the value is always 3. In

the actual flight process, the "Number of Visible Satellites" change is jittered, so a jitter should be added to this feature. In the real situation, the value of the "Number of Visible Satellites" is between 15 and 20 when no one is flying normally, and the value is between 5 and 10 when there is a GPS failure [20].

The function of Equation (3) is used to generate the "Number of Visible Satellites" when the GPS fails and when the GPS works normally. In Equation (3), $x \in [Min_{Sate}, Max_{Sate}]$, $f(x) \in [0, Max_{Sate}]$, where $\mu$ is the parameter controlling the curvature of the curve, $\mu \in (0, \infty)$; the smaller the $\mu$, the gentler the curve, and the larger the $\mu$, the steeper the curve. $Min_{Sate}$ is the minimum number of satellites generated, $Max_{Sate}$ is the maximum number of satellites generated, and $Min_{Sate}, Max_{Sate} \in N^*$.

Take $Min_{Sate}$ as 3 and $Max_{Sate}$ as 10 in the fault state, at this time $x \in [3, 10]$, $f(x) \in [0, 10]$. The Number of Visible Satellites is 0, which means that there is no satellite to provide services for the UAV under the GPS failure state, and the Number of Visible Satellites is 10, which means that the GPS service provider can provide the maximum number of GPS usage in the GPS failure state. Take $Min_{Sate}$ as 8 and $Max_{Sate}$ as 20 in the normal state, at this time $x \in [8, 20]$, $f(x) \in [0, 20]$. The Number of Visible Satellites generated is between 0 and 20, and the number of satellites taken at this time is 8 to 20. The Number of Visible Satellites is 8, which is the number of GPS required to maintain the normal flight of the drone under normal GPS conditions. The Number of Visible Satellites is 20, which is the maximum number of GPS usages that the GPS service provider can provide when the GPS is working normally.

$$f(x) = \frac{Max_{Sate}}{\sqrt{2\pi} * \mu} e^{-\left(\frac{(x - Max_{Sate})^2}{2 * (\mu)^2}\right)} \tag{3}$$

The "Number of Visible Satellites" produced by Equation (3) follows a normal distribution. The graphs produced by Equation (1) under normal and abnormal conditions are shown in Figure 7. The solid orange line represents the image of the "Number of Visible Satellites" generated by when the GPS fails, and the dotted orange line represents the unused value range beyond the specified range. The solid gray line represents the "Number of Visible Satellites" function image generated when the GPS is normal. The gray dotted line represents the unused value range beyond the specified range.

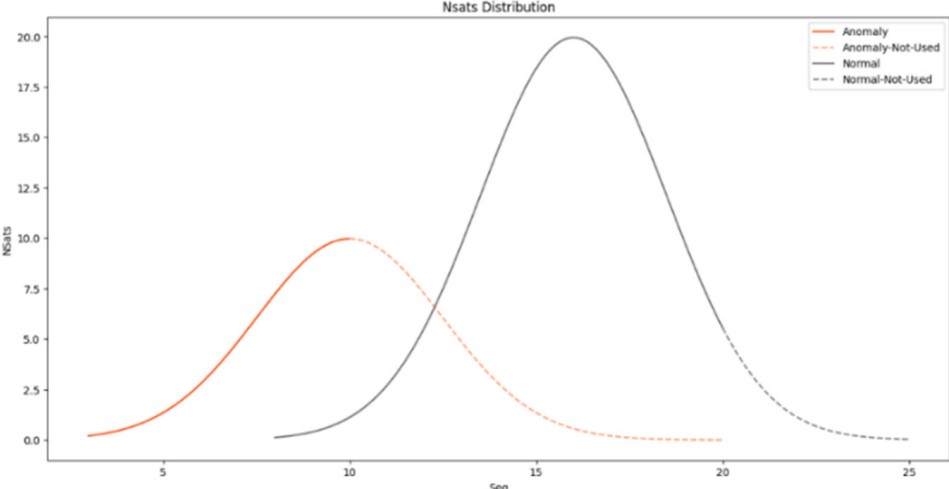

**Figure 7.** $f(x)$ under normal and abnormal GPS conditions.

The change in the "Number of Visible Satellites" before and after adding dithering is shown in Figure 8. As shown in Figure 8a, before adding dithering, the value of "Number of Visible Satellites" is 3 or 10; after adding jitter, it has more abundant values.

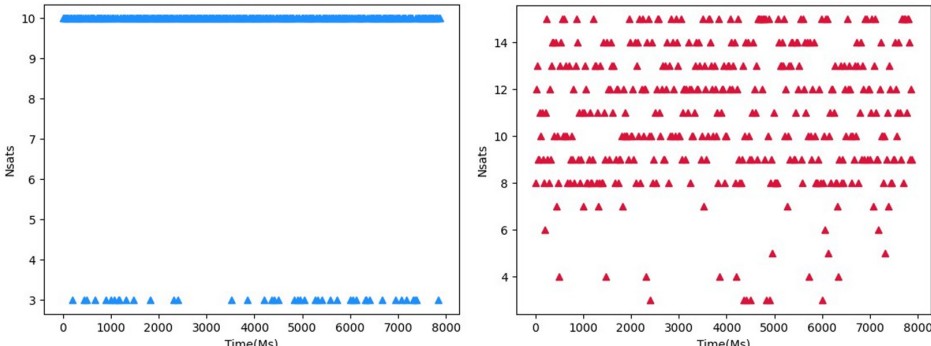

(a) Changes in the number of Nsats before adding dithering  (b) Changes in the number of Nsats after adding dithering

**Figure 8.** The change of "Number of Visible Satellites" before and after adding jitter.

### 3.4. Simulation of GPS Faults

(1)    Simulation parameter configuration when simulating GPS failure

When performing software-in-the-loop simulations of UAVs, the parameters can be changed to simulate the effects of wind speed, wind direction, sensor failure, etc., on UAVs. For example, set "SIM_GPS_DISABLE" from 0 to 1 to simulate a GPS failure during the operation of the drone. Some parameters for simulating UAV failure are shown in Table 1.

**Table 1.** Some parameter settings in the GPS failure simulation.

| Parameter | Value | Description |
| --- | --- | --- |
| SIM_WIND_T_ALT | 60.000000 | Full wind height |
| SIM_GPS_DISABLE | 0/1 | Clear GPS fault/simulate GPS fault |
| FS_EKF_THRESH | 0/1 | Disable/enable fail-safe mechanism |
| SIM_BATT_VOLTAGE | 12.6 | Simulate ambient battery voltage |

(2)    GPS failure simulation

The trajectory of the UAV at the ground station QGroundControl is shown in Figure 9. From point A to B and from point C to D, the drone experienced a GPS failure; from point B to C, the drone's GPS returned to normal. In the simulation environment, after losing the GPS signal, the drone's response is to keep the current running position and continue to fly forward. After the GPS signal is restored, the drone will move toward the received GPS coordinates at a speed different from normal flight until it reaches the designated waypoint.

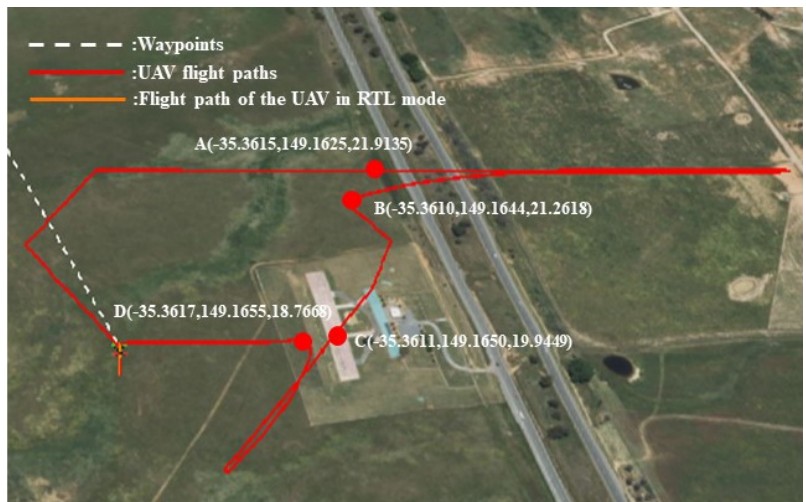

**Figure 9.** The flight trajectory of the UAV in the state of GPS failure.

In order to understand the impact of GPS failure on the drone, we extracted the drone's information from the log. Figure 10a shows that after the GPS failure, the UAV stalled, and the UAV ran at a relatively fast speed when the TimeStamp is 500,000 ms. When the TimeStamp was 600,000 ms, the GPS failure of the UAV recovered, and there was a short pause. Since the flight mission had to be continued, it returned to the original waypoint at a relatively fast speed. From Figure 10b,c, due to the change of velocity, the Circular Angle and acceleration were abnormal. It can be seen from Figure 10d that after the GPS failure, the UAV lost its accurate judgment on its position, so the roll angle and pitch angle of the UAV were abnormal.

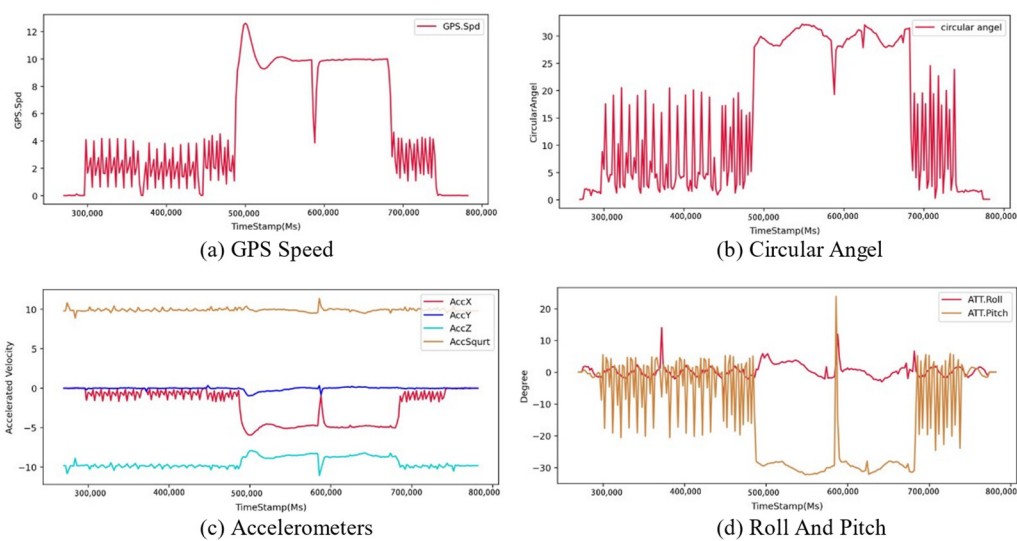

(a) GPS Speed    (b) Circular Angel

(c) Accelerometers    (d) Roll And Pitch

**Figure 10.** The status change of the drone when GPS fails.

### 3.5. Simulation of Accelerometer Faults

(1)    Simulation parameter configuration when simulating the accelerometer failure

Both accelerometer one and accelerometer two were disabled in the simulation environment by setting SIM_ACCEL1_FAIL and SIM_ACCEL2_FAIL to 1. Since the drone cannot acquire correct acceleration information, it cannot derive a current, correct position estimate and heading estimate to simulate accelerometer failure. Some parameters of the simulation environment when simulating Accelerometer failure are shown in Table 2.

**Table 2.** Some parameter settings in accelerometer fault simulation.

| Parameter | Value | Description |
| --- | --- | --- |
| SIM_WIND_T_ALT | 60.000000 | Full wind height |
| FS_EKF_THRESH | 0/1 | Disable/enable fail-safe mechanism |
| SIM_ACCEL1_FAIL | 0/1 | Clear/simulate accelerometer 1 fault |
| SIM_ACCEL2_FAIL | 0/1 | Clear/simulate accelerometer 2 fault |

(2)    The Simulation of Accelerometer Failure

By changing the parameters of the accelerometer during the UAV flight, the UAV accelerometer's failure state was simulated. The trajectory of the UAV under QGround-Control is shown in Figure 11. Accelerometer failures occurred from point A to point B and from point C to point D. It can be seen that after the drone fail-safe mechanism was disabled and the drone lost acceleration information, the flight trajectory presented an unstable state. The rest of the flight track accelerometers were working normally, and the flight track of the UAV presented a stable state.

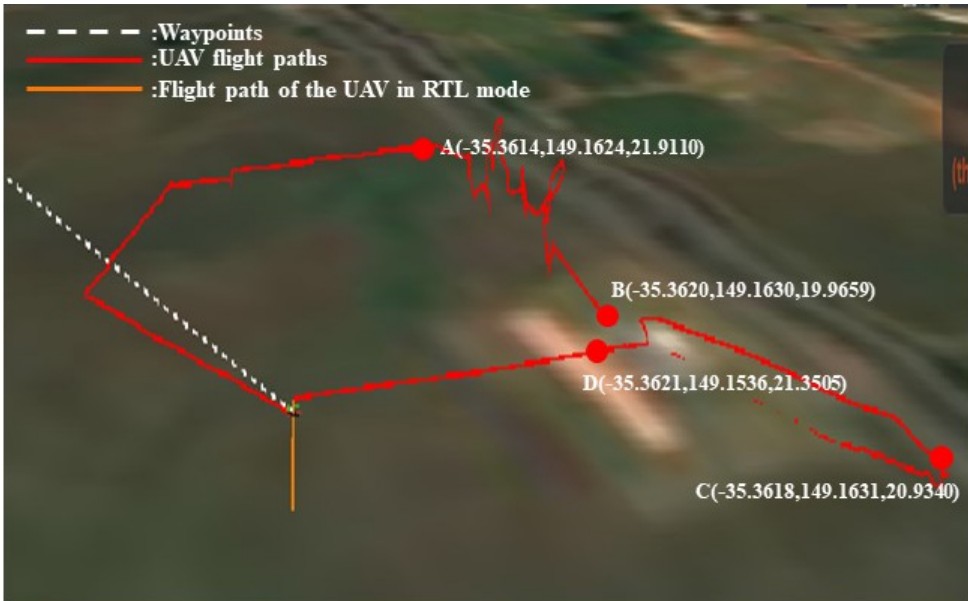

**Figure 11.** The flight trajectory of the UAV in the state of accelerometer failure.

In order to better understand the impact of accelerometer failure on the UAV, the status information of the UAV, as shown in Figure 12, was extracted after the flight. As shown in Figure 12a, the accelerometer failure occurred when the TimeStamp was about 250,000 Ms to 390,000 Ms, and the GPS speed of the drone had abnormal vibrations. The reason is that after the drone lost accurate position estimation after the accelerometer failure, the UAV tried to maintain the current flight trajectory through past position information so that the speed will be unstable. Due to the change in speed, as shown in Figure 12b, the UAV's Circular Angle also appeared abnormal. As shown in Figure 12c, after the accelerometer failure, the acceleration information of the drone in the x, y, and z axes were completely lost. Since accurate position estimation could not be obtained through the acceleration information, so the roll angle and pitch angle were also abnormal during this period.

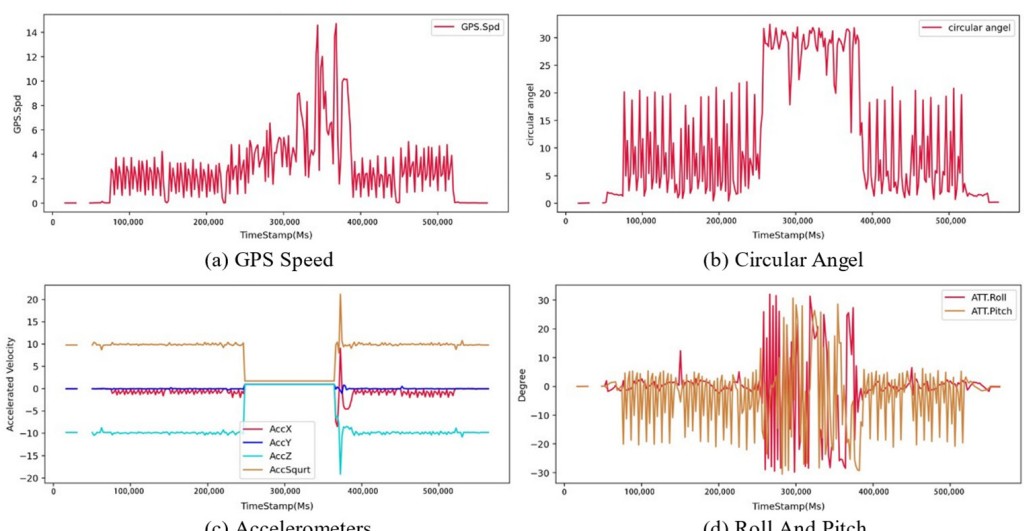

**Figure 12.** State change of drone with accelerometer failure.

### 3.6. Simulation of Engine Faults

(1)  Simulation parameter configuration when simulating engine failure

In the simulation environment, SIM_ENGINE_MUL was set to 0 to simulate the failure of the drone's engine. Since parameters such as wind force and wind speed have little effect on engine failure, the wind speed is always 0 during the experiment. Some parameters of the simulation environment when simulating engine failure are shown in Table 3.

**Table 3.** Parameter settings in the engine failure simulation.

| Parameter | Value | Description |
| --- | --- | --- |
| SIM_WIND_T_ALT | 60.000000 | Full wind height |
| FS_EKF_THRESH | 0/1 | Disable/enable fail-safe mechanism |
| SIM_ENGINE_MUL | 1/0 | Clear/simulate engine failure |
| SIM_BATT_VOLTAGE | 12.6 | Simulate ambient battery voltage |

(2)  The Simulation of Accelerometer Fail

The running trajectory of the UAV under QGroundControl is shown in Figure 13. Engine failure occurred from point A to point B and from point C to point D. In the simulation environment when the drone's engine failed, the drone fell from the air to the ground. If the fault is recoverable, the UAV can be launched from the ground to perform the flight mission again. The UAV will quickly take off from the ground to continue the mission.

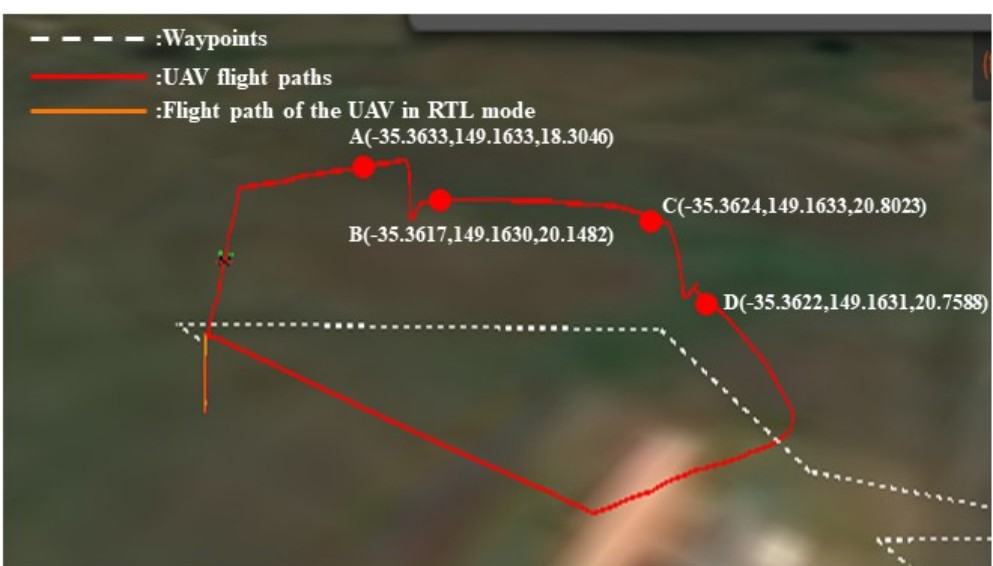

**Figure 13.** The flight trajectory of the UAV in the state of engine power loss.

In order to understand the impact of power loss on the state of the UAV, this article extracts some information for analysis. As shown in Figure 14a, when the TimeStamp was about 330,000 Ms, the power of the UAV was lost, and its GPS speed was 0. After the power was restored, the UAV took off quickly and continued to perform flight missions. From Figure 14b,c, when the TimeStamp was about 320,000 Ms, before the UAV's power was lost, its circular surface angle and acceleration were abnormal and then became 0. As shown in Figure 14d, when the TimeStamp was about 330,000 Ms, the drone's pitch and roll angles were abnormal and became 0 after landing. That is because the UAV fell to the ground from a high altitude in free fall after losing power, and its roll angle showed irregular changes. After landing, the UAV was still, so the angle did not change.

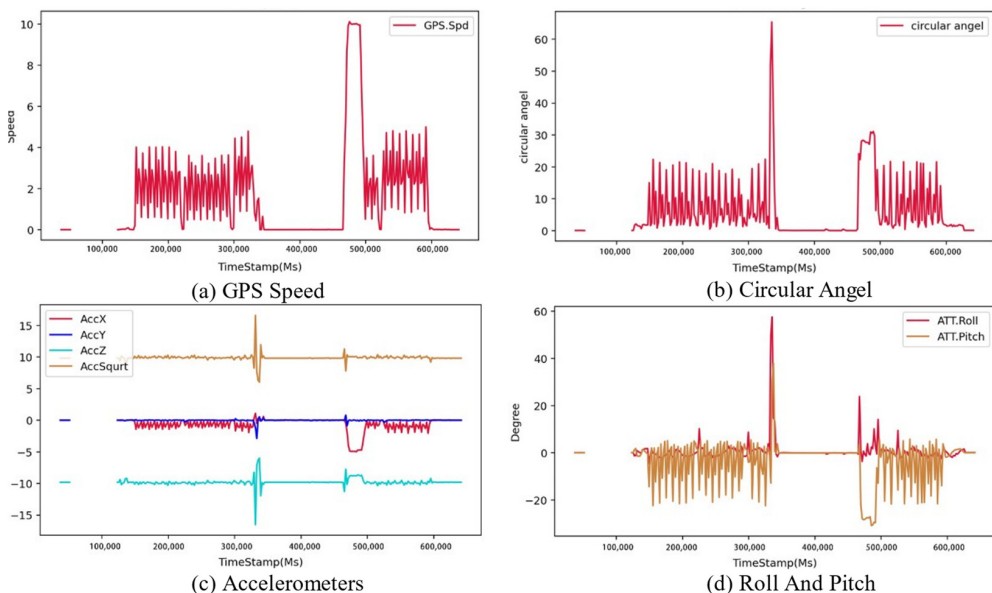

**Figure 14.** UAV state changes when engine power is lost.

*3.7. Simulation of Remote-Control System Faults*

(1)     Simulation parameter settings when simulating remote control failure

A remote controller is a tool for the operator to control the drone and send instructions to the drone. When the parameter SIM_RC_FAIL is set to 1, it means that the simulated Remote-Control System failure starts. Table 4 lists some parameters of the simulation environment when simulating a remote controller failure.

**Table 4.** Parameter settings in the Remote-Control System failure simulation.

| Parameter | Value | Description |
| --- | --- | --- |
| FS_EKF_THRESH | 0/1 | Disable/enable fail-safe mechanism |
| SIM_RC_FAIL | 0/1 | Clear/simulate remote control faults |
| SIM_BATT_VOLTAGE | 12.6 | Simulate ambient battery voltage |

(2)     The Simulation of Remote-Control System Failure

In the event of a remote controller failure, the UAV loses all communication with the remote controller. The UAV starts the RTL (Return to Launch) mode, ends the current mission, and returns to the take-off point immediately. However, the UAV in a normal state will completely perform the originally set flight mission. The planned flight trajectory of the UAV is shown in Figure 15a,b, and the flight trajectory when the remote controller fails is shown in Figure 15c,d. There was a remote failure between points A and B.

In order to better understand the change in the state of the UAV when the remote controller fails, some information about the UAV is extracted after the flight. As shown in Figure 16a, the remote-control failure occurred when the TimeStamp was about 350,000 Ms. Since the UAV directly activated the RTL mode, its acceleration differed from that of a normal flight. As shown in Figure 16b, when the UAV was in RTL mode, it did not turn the course of the UAV but returned directly, so its pitch angle and roll angle was abnormal when the TimeStamp was about 350,000 Ms.

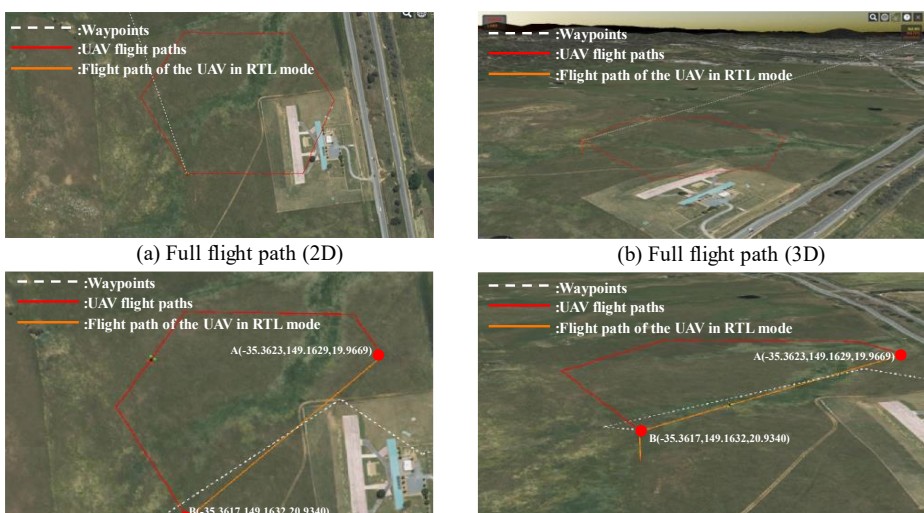

(a) Full flight path (2D)  (b) Full flight path (3D)

(c) Incomplete flight path when remote control fails (2D) (d) Incomplete flight path when remote control fails (3D)

**Figure 15.** The flight trajectory of the UAV in the normal state and the fault state.

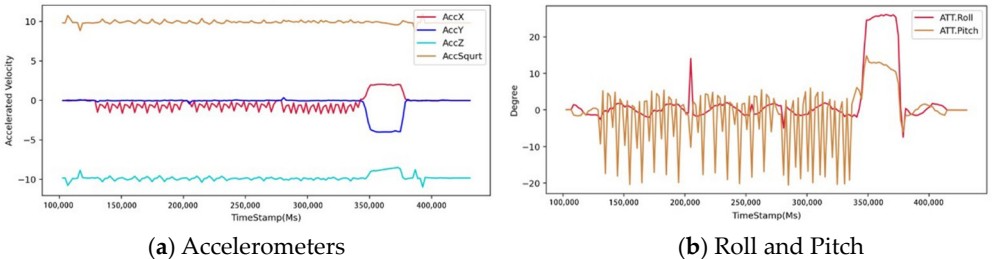

(**a**) Accelerometers  (**b**) Roll and Pitch

**Figure 16.** UAV status changes when the remote controller fails.

## 4. Processing of UAV Fault Data

### 4.1. The Introduction of UAV Fault Data and the Content of This Section

The UAV fault data refers to the on-board logs that the drone records when a component of the drone fails. As can be seen from the previous section, this paper simulates GPS failure, accelerometer failure, engine failure, and remote-control failure in the same flight. Additionally, it realizes the original collection of on-board logs when the drone fails. This section introduces the processing method of UAV fault data proposed in this paper includes the Failure Time Anchoring method and Time Window Stretching method.

### 4.2. Convert Log to CSV

Since the drone's log is recorded as a log, it cannot be directly processed by the method based on artificial intelligence, so the log file needs to be converted. The Mission Planner (MP) (https://ardupilot.org/planner/, accessed on 13 December 2022) ground station can load and store the drone's log as a matrix file. After processing, it can be converted into a CSV file. This article extracted four different sensor information, and the specific information names and their corresponding physical meanings are shown in Table 5.

**Table 5.** Extracted sensor information and names.

| Name | Physical Meaning | Data Size |
|------|------------------|-----------|
| GPS | Information received from GNSS attached to autopilot | $15 \times 2450$ |
| IMU | Inertial Measurement Unit Data | $15 \times 12253$ |
| RATE | Desired and achieved vehicle attitude rates | $13 \times 4900$ |
| VIBE | Processed (acceleration) vibration information | $6 \times 4900$ |

### 4.3. Anomalies Time Window Setting Based on Fault Time Point Anchoring Method

The experimental method was adopted for simulating GPS, accelerometer, engine (recoverable), and remote controller failures during one flight. There should be a reasonable division method to distinguish normal data from abnormal data. The method used in this paper is as follows: when the fault occurs, record it as $T_{start\_i}$ and record the longitude, latitude, and altitude at this time. After the flight, go to the GPS information recorded in the log to find the longitude, latitude, and altitude when the fault occurred so as to determine the starting position of the faulty sample. Similarly, when the fault ends, record it as $T_{end\_i}$, the longitude, latitude, and altitude at this time are also recorded to determine the location of the sample at the end of the fault. The mathematical description is shown in Equation (4). Table 6 shows the correspondence between the fault types represented by each class label and the fault window.

$$\begin{cases} T_{start_i} = Record(T \,|\, \{Lat_k, Lon_k, Alt_k\}) \\ T_{end_i} = Record\big(T \,|\, \{Lat_j, Lon_j, Alt_j\}\big) \quad i \in [0, FlyTimes] \\ TimeWindow_i = T_{end\_i} - T_{start\_i} = Fault_i \end{cases} \tag{4}$$

**Table 6.** Fault types represented by class labels.

| Labels | Fault Type | Window Type |
|:------:|:----------:|:-----------:|
| 0 | Normal | Fault0 |
| 1 | GPS failure | Fault1 |
| 2 | Accelerometer failure | Fault2 |
| 3 | Engine failure | Fault3 |
| 4 | Remote-Control System failure | Fault4 |

Among them, $Lat_k$ and $Lat_j$ refer to the latitude of the drone at a certain moment; $Lon_k$ and $Lon_j$ refer to the longitude of the drone at a certain moment; and $Alt_k$ and $Alt_j$ refer to the altitude of the drone at a certain moment. $Record()$ refers to the function of recording the longitude, latitude, and altitude of the drone. $TimeWindow_i$ refers to the time window of the *i*-th fault. $FlyTimes$ refers to the number of failed flights/normal flights.

### 4.4. Feature Processing

The drone's log contains many features, but some are not universal, such as changes in the longitude and latitude over a certain period because during the training process it is impossible to simply determine that a UAV has malfunctioned based on its arrival at a particular latitude and longitude. Moreover, the time-dependent feature should be removed, as it only serves to mark the sample location. The feature processing method is shown in Algorithm 2.

---

**Algorithm 2:** Feature Processing

---

Input: Single CSV file with original features
Output: CSV file with processed features
for *Feature_i* in CSV:
   if *Feature_i* Related to time || *Feature_i* Not universal:
     Delete (*Feature_i*);
end if;
end for;
Merge the remaining *Feature* into one CSV;
end;

---

*4.5. Data Balance Processing Based on Failure Time Window Stretching Method*

(1)  Data distribution before time window stretching

The number of instances per file for the completed merged CSV files is too small. For example, there are only 2555 instances of GPS information, and the specific data distribution is shown in Table 7. The small size of the sample data is not conducive to learning based on artificial intelligence models. Additionally, there is an imbalance between normal and abnormal data, as shown in Figure 17; the normal data comprised 74.21%, while the remaining abnormal data accounts for only 25.79%. The imbalance between positive and negative samples results in under-learning for negative samples for the model.

**Table 7.** Data distribution table for data before balancing treatment.

|  | **Normal** | **GPS Fail** | **Accelerometer Fail** | **Engine Fail** | **RC Fail** | **Total** |
|---|---|---|---|---|---|---|
| GPS | 1896 | 129 | 108 | 104 | 318 | 2555 |
| IMU | 9479 | 645 | 539 | 519 | 1591 | 12,773 |
| RATE | 3794 | 258 | 216 | 208 | 633 | 5109 |
| VIBE | 3793 | 258 | 216 | 208 | 634 | 5109 |

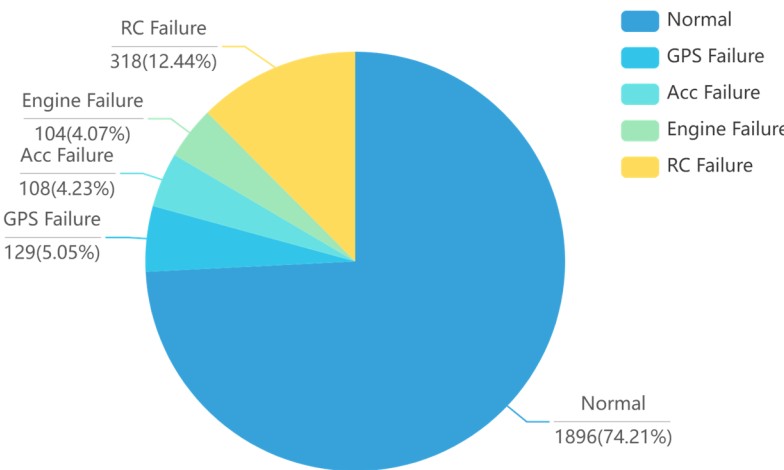

**Figure 17.** Percentage of labels before data processing.

(2)  Data balancing methods

In this paper, the data is balanced by the method of Fault Time Window Stretching, and the time window of the original normal flight is defined as $Time_{Normal}$, and the time window for the abnormal flight is $Time_{Fault}$, $Time_{Normal}$, $Time_{Fault} \neq 0$. The amount of data per unit of time is $\beta$, $\beta \epsilon R^*$, the ratio of the abnormal data volume to the total data volume is $\alpha$, $\alpha \epsilon [0,1]$, and $\alpha$ is calculated as shown in Equation (5). The ratio of target abnormal data to all data is $\gamma$, $\gamma \in [0,1]$. $\gamma$ is calculated as shown in Equation (5) where $\lambda$, $\lambda \in R$ is the failure time window control factor. The duration of the fault window can be increased when $\lambda > 1$, thus increasing the amount of fault data in one flight; the opposite is true when $\lambda < 1$.

$$\begin{cases} \alpha = \frac{\beta * Time_{Fault}}{Time_{Normal} + Time_{Fault}} \\ \gamma = \alpha * \lambda = \frac{(\beta * Time_{Fault})}{Time_{Normal} + Time_{Fault}} * \lambda \end{cases} \quad (5)$$

As shown in Table 7, taking the GPS information as an example, before the balancing process, the number of normal data is 1896, the number of fault data is 659, and the data of fault types account for 25.79% of all data. In order to facilitate the processing, the data obtained from the four flights were combined. Figure 18 shows the proportion of each label after processing. As shown in Table 8, after the balancing process and combining process,

the number of normal instances is 3609, the number of all fault type instances is 4284, and the data of faults accounts for 54.27% of all data, which increases the fault data by 28.48%. An increase in the number of abnormal samples will facilitate the learning of abnormal data by the AI-based model and improve the model's generalization.

**Figure 18.** The proportion of each label after processing.

**Table 8.** Data distribution table of the data set after the balancing process.

|       | Normal | GPS Fail | Accelerometer Fail | Engine Fail | RC Fail | Total  |
|-------|--------|----------|--------------------|-------------|---------|--------|
| GPS   | 3609   | 1096     | 1305               | 606         | 1277    | 7893   |
| IMU   | 18,032 | 5480     | 6528               | 3033        | 6393    | 39,466 |
| RATE  | 7215   | 2188     | 2610               | 1215        | 2558    | 15,786 |
| VIBE  | 7218   | 2189     | 2612               | 1210        | 2557    | 15,786 |

## 5. Experimentation

In order to demonstrate the validity of the data set, ten classical machine learning algorithms, as well as a convolutional neural network model (CNN), were used to evaluate the data set in this paper. The evaluation results are expressed as the precision, recall, and F-Measure of each machine learning algorithm using this data set. Four sensor information, namely GPS information, Inertial Measurement Unit (IMU), desired and achieved vehicle attitude rate (RATE), and processed acceleration vibration information (VIBE), were selected as data sets. Before feeding the data into the model training, the feature processing method of Algorithm 2 was applied to each information data to filter the features about time as well as the features that are not generic.

In the data set, sensors or telemetry information are recorded at different scales, which leads to some features whose values are not in the same order of magnitude, which can lead to an imbalance in linear operations and can affect the results of machine learning. The $Max - Min$ feature scaling method was used to scale each feature between 0 and 1, using the method shown in Equation (6).

$$Feature_{scaled} = \frac{Feature_i - Min(Feature_i)}{Max(Feature_i) - Min(Feature_i)} \tag{6}$$

Precision can be understood as the ratio between the number of detected features of a certain class and the number of all detected features; Recall can be understood as the ratio between the number of detected features of a certain class and the number of all features of that class in the data set. Since Precision and Recall values sometimes appear

contradictory, they need to be considered together, and F-Measure is the weighted summed average of Precision and Recall. Precision, Recall, and F-Measure are calculated as shown in Equations (7) and (8).

$$Precision = \frac{TP}{(TP+FP)} \tag{7}$$

$$Recall = \frac{TP}{(TP+FN)} \tag{8}$$

In Equations (7) and (8), TP refers to the number of positive samples correctly identified, FP refers to the number of negative samples that are misreported, and FN refers to the number of negative samples predicted by the classifier that is actually positive.

$$F_1 Measure = \frac{2 * Precision * Recall}{Precision+Recall} \tag{9}$$

In Equation (9), Precision is the calculation result of Equation (7) and Recall is the calculation result of Equation (8).

### 5.1. Comparison with Related Data Sets

#### 5.1.1. Data Information Comparison

Table 9 highlights the significant shortage of data sets available for UAV fault detection at present. Moreover, many of the available data sets have imbalanced positive and negative samples, making them inadequate for effectively training artificial intelligence-based anomaly detection models for UAVs. In this context, the proposed TLM method offers a viable solution for obtaining UAV fault data, as well as facilitating the labeling and balancing of positive and negative samples.

**Table 9.** Comparison of different UAV anomaly detection data sets.

| Data Sets | Year | Number of Features | Abnormal Number | The Proportion of Abnormal Samples (%) | Labelled |
|---|---|---|---|---|---|
| UAV ATTACK [21] | 2021 | 1110 | 2 | 2.2 | No |
| ECU-IoFT [15] | 2022 | 10 | 3 | 39.0 | Yes |
| TLM(Ours) | 2023 | 821 | 4 | 54.3 | Yes |

#### 5.1.2. Performance Comparison

As the UAV ATTACK data set does not contain clear positive and negative sample labels, this paper utilized the K-Means algorithm to cluster the GPS information of UAV AT-TACK and generated appropriate labels. The labeled GPS information was then subjected to experimentation using the KNN algorithm to determine its accuracy rate. Notably, the accuracy of KNN on the ECU-IoFT data set was obtained from [15]. The results presented in Table 10 demonstrate that KNN achieved the highest accuracy rate on the proposed data set in this paper.

**Table 10.** Accuracy of using KNN algorithm on different data sets.

| Data Sets | Accuracy Using KNN Algorithm (%) |
|---|---|
| UAV ATTACK | 47.76 |
| ECU-IoFT | 21.42 |
| TLM(Ours) | 70.65 |

### 5.2. Performance on Machine Learning Algorithm

From Figure 19, it can be seen that the Random Forest algorithm achieved a high accuracy rate of 84.0851% when using GPS information as data for training. The tree algorithm was influenced by the key features, and the changes in several of its features were obvious when the UAV fails, so the tree algorithm had a higher accuracy rate. The

worst performing algorithm was the ZeroR, with an accuracy of 47.8966%. ZeroR is a simple classifier that selects a category with the highest probability as the classification result of an unknown sample. The performance of the other algorithms was concentrated between 50% and 70%, and better classification algorithms are needed to classify this data set so as to accurately identify the normal and abnormal states of drones by analyzing information from multiple aspects.

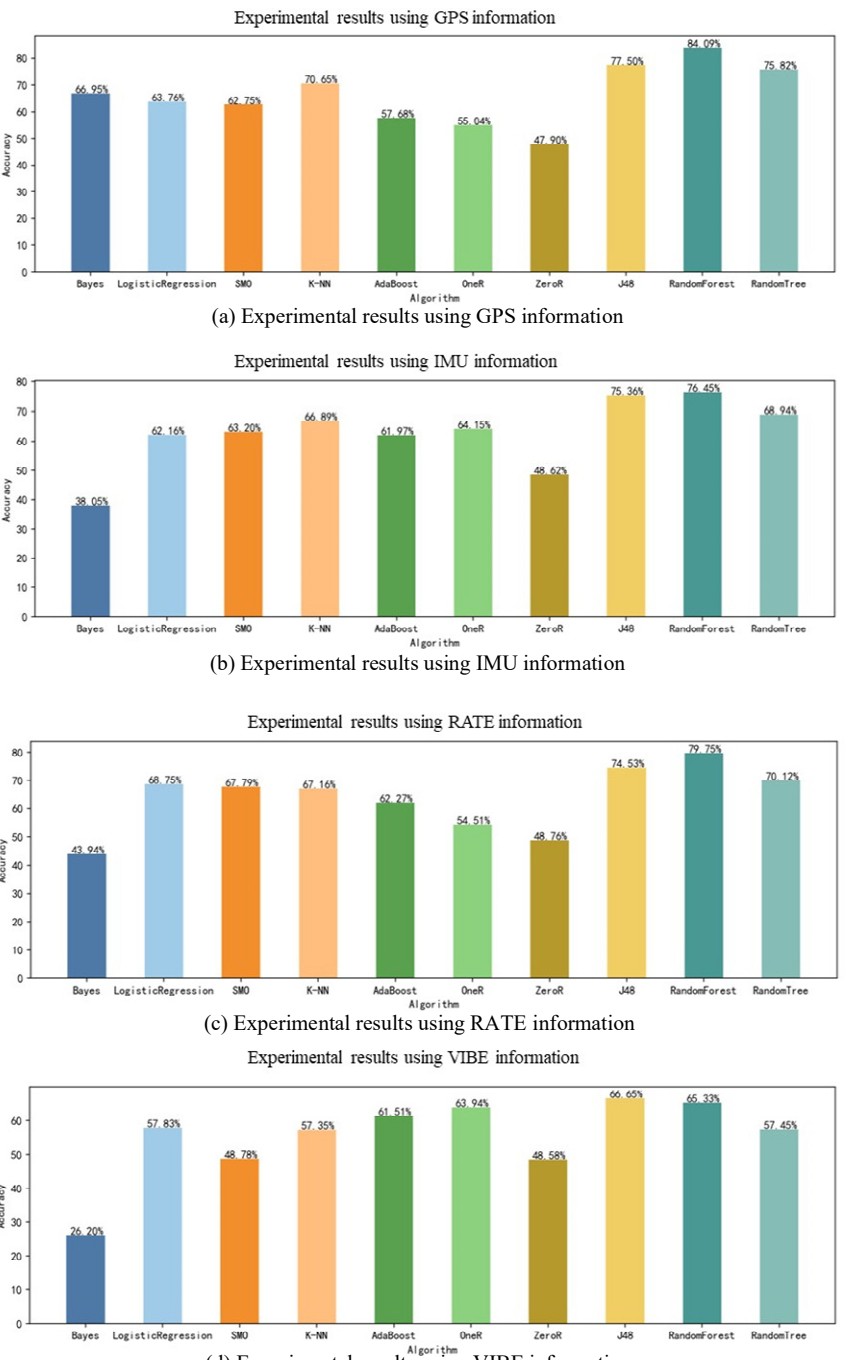

(a) Experimental results using GPS information

(b) Experimental results using IMU information

(c) Experimental results using RATE information

(d) Experimental results using VIBE information

**Figure 19.** Experimental Results.

The results of the evaluation of the data set using other classification metrics are shown in Table 9; the highest accuracy rate is indicated by bolded text. As seen from the table, the accuracy of the tree algorithm was generally higher than that of the others. The algorithm for tree categories relies on just a few features. However, it is often incorrect to judge

abnormal or normal based on a certain characteristic in real situations. For example, in the event of a battery failure in a UAV, there will be a sudden increase in discharge during a certain period. However, imagine the following scenario, in which the UAV encounters strong wind resistance, the UAV will increase its speed to prevent the mission path from deviating, and the battery will increase its discharge to keep the engine running normally, but the UAV does not have a battery failure at this time.

In addition, as shown in Table 11, the accuracy of SMO, AdaBoost, and ZeroR algorithms is NaN because the denominator is 0 in the calculation of Precision. According to the formula of Precision, i.e., TP + FP is 0, which means that none of the samples were correctly classified. That also indicates that the classification of this data set is complex, and traditional machine learning algorithms cannot accurately classify faults.

**Table 11.** Evaluation of multi-classification algorithms using GPS data.

| Method | Correct Classified% | Incorrect Classified% | TP | FP | Precision | Recall | F1 |
|---|---|---|---|---|---|---|---|
| Naïve Bayes | 66.95% | 33.04% | 0.670 | 0.205 | 0.758 | 0.670 | 0.646 |
| Logistics | 63.76% | 36.24% | 0.638 | 0.265 | 0.696 | 0.638 | 0.559 |
| SMO | 62.75% | 37.25% | 0.627 | 0.275 | NaN | 0.627 | NaN |
| KNN | 70.65% | 29.34% | 0.707 | 0.122 | 0.733 | 0.707 | 0.709 |
| AdaBoost | 57.68% | 42.32% | 0.577 | 0.389 | NaN | 0.577 | NaN |
| OneR | 55.04% | 44.96% | 0.550 | 0.287 | 0.497 | 0.550 | 0.511 |
| ZeroR | 47.89% | 52.10% | 0.479 | 0.479 | NaN | 0.479 | NaN |
| J48 | 77.49% | 22.50% | 0.775 | 0.121 | 0.787 | 0.775 | 0.774 |
| RandomForest | **84.08%** | 15.91% | 0.841 | 0.081 | 0.851 | 0.841 | 0.840 |
| RandomTree | 75.82% | 24.17% | 0.758 | 0.126 | 0.766 | 0.758 | 0.753 |

As shown in Tables 12 and 13, the random forest algorithm and the Naive Bayesian algorithm are selected in this paper, and the classification effect was evaluated using a confusion matrix, from which it can be seen that the classification of the accelerometer faults was the most difficult. In the Naive Bayesian and the random forest algorithm, 288 and 122 normal data were incorrectly classified as accelerometer faults, respectively.

**Table 12.** Confusion matrix of Naive Bayes algorithm.

| Normal | GPS | Acc | Engine | RC | Class |
|---|---|---|---|---|---|
| 780 | 12 | 17 | 134 | 2 | Normal |
| 0 | 193 | 0 | 0 | 0 | GPS |
| 288 | 3 | 153 | 0 | 1 | Acc |
| 1 | 0 | 0 | 126 | 0 | Engine |
| 128 | 0 | 0 | 66 | 69 | RC |

**Table 13.** Confusion matrix of random forest algorithm.

| Normal | GPS | Acc | Engine | RC | Class |
|---|---|---|---|---|---|
| 847 | 3 | 56 | 35 | 4 | Normal |
| 0 | 193 | 0 | 0 | 0 | GPS |
| **112** | 0 | 293 | 17 | 23 | Acc |
| 2 | 0 | 0 | 125 | 0 | Engine |
| 31 | 0 | 0 | 31 | 201 | RC |

*5.3. Performance on Convolutional Neural Networks*

In order to test the performance of the data set obtained by the TLM method on the neural network, the information of GPS, IMU, RATE, and VIBE were extracted and input into the CNN for training. This is done by converting the data with 16-dimensional features

and 9-dimensional features into grayscale maps and inputting them into CNN for training. The accuracy and loss during training are shown in Figures 20 and 21.

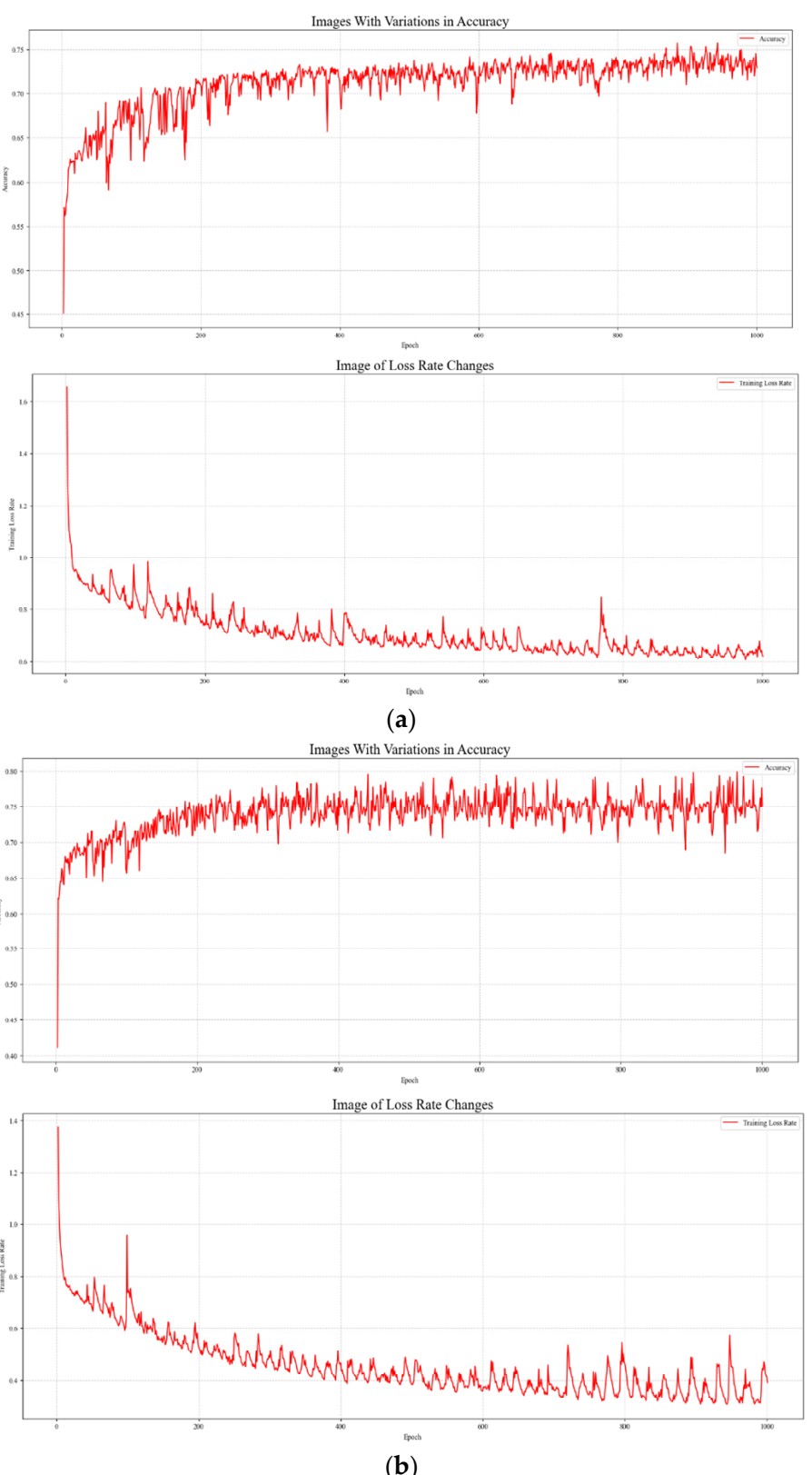

**Figure 20.** (**a**) Training accuracy and loss change of IMU. (**b**) Training accuracy and loss change of RATE.

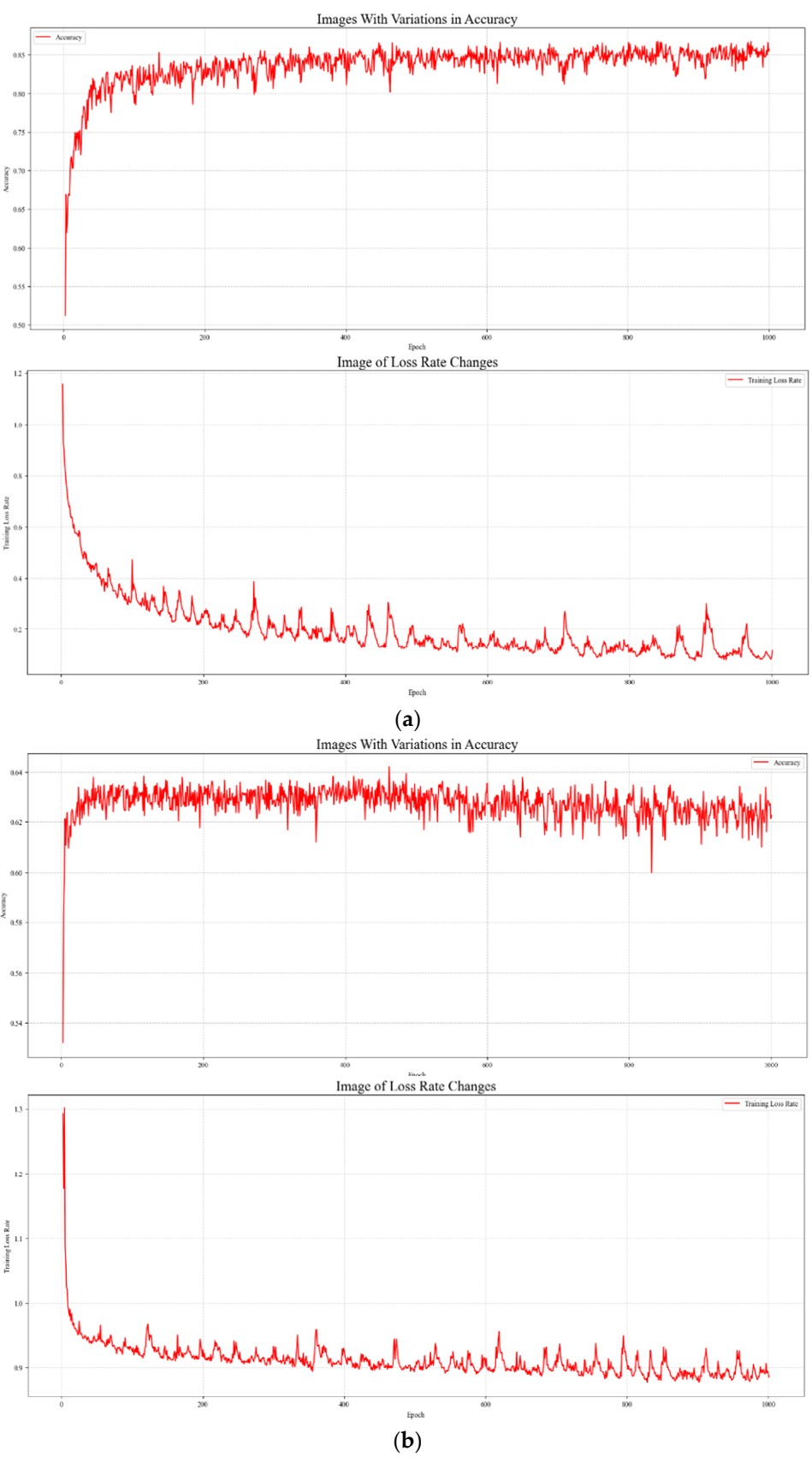

**Figure 21.** (**a**) Training accuracy and loss change of GPS. (**b**) Training accuracy and loss change of VIBE.

From Figure 20a,b, it can be seen that the accuracy of IMU and RATE only converged to 75% during training; from Figure 21a, the accuracy of GPS information converged to about 85% during training. From Figure 21b, the accuracy of VIBE information converged to 63% during training. The test accuracies on the test set are shown in Table 14. It can be seen that the GPS data had the highest classification accuracy of 85.63%. Comparing other machine learning algorithms with CNN, the accuracy of CNN did not have a significant advantage on this task.

**Table 14.** Test results from CNN.

| Category | Accuracy |
| --- | --- |
| GPS data | **85.63%** |
| RATE data | 75.32% |
| IMU data | 73.03% |
| VIBE data | 63.15% |

Through analysis, the gap between normal and abnormal samples was too small; after converting the data to images, the differences between positive and negative samples were further reduced, making the learning of CNN using images difficult. In this paper, two samples from different failures were selected to illustrate the above issues, and their grayscale plots are shown in Figure 22.

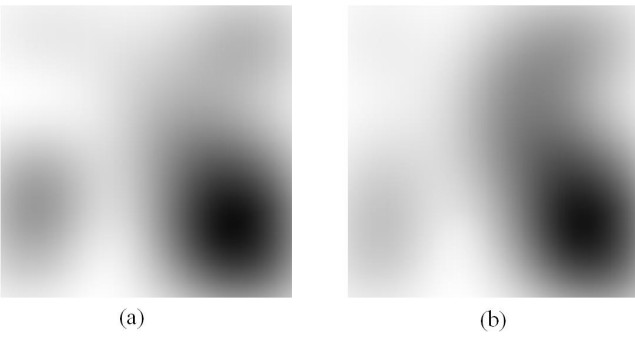

(a)                                        (b)

**Figure 22.** After converting the data into a grayscale image, the comparison between (**a**) normal sample and (**b**) abnormal sample.

Tables 15–18 show the classification results of the different data obtained from UAVs on CNN, Normal, GPS, Acc, Engine, and RC, referring to the four types of fault classification and normal classification, respectively. In Table 13, which shows the results of testing using the GPS data, the highest classification accuracy of 96.32% was achieved for GPS faults. That is because the differences between the individual features in the GPS information are larger, so it is beneficial to the training of the model. In Table 16, which shows the classification effect using VIBE vibration data, the classification accuracy for Acc fault was 0. By analysis, it was found that in VIBE information, there were many samples duplicated at the Acc fault; there is no change in vibration information at the accelerometer fault, so the features were not learned. The classification effect of the remaining information was between 60% and 80%.

**Table 15.** Results of GPS data.

| | F1-Score | Accuracy | Recall |
| --- | --- | --- | --- |
| Normal | 0.8881 | 0.8846 | 0.8916 |
| GPS | 0.9587 | 0.9632 | 0.9544 |
| Acc | 0.8436 | 0.8792 | 0.8108 |
| Engine | 0.6269 | 0.6802 | 0.5813 |
| RC | 0.7939 | 0.7470 | 0.8471 |

**Table 16.** Results of IMU data.

|  | **F1-Score** | **Accuracy** | **Recall** |
|---|---|---|---|
| Normal | 0.7552 | 0.7325 | 0.7793 |
| GPS | 0.6271 | 0.7551 | 0.5363 |
| Acc | 0.8360 | 0.8364 | 0.8355 |
| Engine | 0.6790 | 0.5516 | 0.8829 |
| RC | 0.6510 | 0.7395 | 0.5814 |

**Table 17.** Results of RATE data.

|  | **F1-Score** | **Accuracy** | **Recall** |
|---|---|---|---|
| Normal | 0.7506 | 0.7186 | 0.7854 |
| GPS | 0.6459 | 0.7560 | 0.5639 |
| Acc | 0.8240 | 0.8687 | 0.7837 |
| Engine | 0.9182 | 0.9207 | 0.9157 |
| RC | 0.6959 | 0.6761 | 0.7169 |

**Table 18.** Results of VIBE data.

|  | **F1-Score** | **Accuracy** | **Recall** |
|---|---|---|---|
| Normal | 0.7158 | 0.6010 | 0.8849 |
| GPS | 0.4796 | 0.6666 | 0.3746 |
| Acc | 0.0000 | 0.0000 | 0.0000 |
| Engine | 0.8919 | 0.9512 | 0.8395 |
| RC | 0.3177 | 0.5622 | 0.2214 |

## 6. Conclusions

UAVs are increasingly used in many industries, so fault detection and anomaly detection for UAVs are becoming increasingly important. When using machine learning and deep learning methods for fault detection of UAVs, problems arise because the logs of UAVs are difficult to obtain, features are difficult to extract, and processes and logs of UAVs are not directly usable. In this paper, a Time Line Modeling (TLM) approach was proposed to acquire and process the failure data of UAVs. First, based on the software-in-the-loop (SITL) simulation system of UAVs, a flight mission with four trajectories is constructed. The four common faults of UAVs (GPS fault, accelerometer fault, engine fault, and Remote-Control System fault) were simulated in the same flight, and the original logs of the UAV flight were collected. Secondly, the Time Line Modeling approach is specifically divided into two stages. The first stage locates the normal and abnormal data of the UAV using a Fault Point-in-time Anchoring-based approach. The second stage expands the abnormal data of the UAV using a Time-Window Stretching-based approach. Finally, the flight logs under four flight trajectories: GPS information, IMU information, RATE information, and VIBE are used as data. The time-related features, as well as non-generic features, are removed and put into the machine learning model and convolutional neural network model for training. The experimental results proved that the UAV fault data set obtained based on TLM methods was effective. In summary, this paper implemented the collection and processing of UAV fault logs using the proposed TLM method. This approach provided a new source of fault data for classifying UAV faults. Our future research will focus on developing effective models for detecting UAV faults based on the data set obtained through the TLM method.

**Author Contributions:** T.Y. and H.D. contributed to the conception of the study; Y.L. performed the experiment and wrote the manuscript; J.C. and X.T. helped perform the experiment and provided constructive discussions. All authors have read and agreed to the published version of the manuscript.

**Funding:** This work was supported by the Sichuan Science and Technology Program (Grant No. 2022YFG0322), the China Scholarship Council Program (Nos. 202001010001 and 202101010003), the Innovation Team Funds of China West Normal University (No. KCXTD2022-3), the Nanchong Federation of Social Science Associations Program (Grant No. NC22C280), and China West Normal University 2022 University-level College Student Innovation and Entrepreneurship Training Program Project (Grant No. CXCY2022285).

**Data Availability Statement:** Not applicable.

**Acknowledgments:** Thanks to everyone who contributed to this work.

**Conflicts of Interest:** The authors declare no conflict of interest.

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
