# Peer review of "Acquisition and Processing of UAV Fault Data Based on Time Line Modeling Method"

_applsci, doi:10.3390/app13074301_

Round 1

Reviewer 1 Report

In conclusion, the work is interesting, but the presentation of the material is incomplete. 

A description of the conditions under which the experiment is carried out is missing. It is also unclear how the model of the UAV and integrated controller and sensors used in the experiment corresponds to the real UAV.

Inaccuracies observed in the work:

- It is not explained what SMO (line 30) and SVM (line 94) are

 -Incorrectly provided references to sources (lines 39, 41, 48, 94, 98, 100, 105, 111, 112, 114, 121, 124, 127, 132, 135, 139, 190).

- Figure 2 lacks detail. I recommend adding information on WP1, WP2, etc, and distance information between WP.

- Figure 5 is difficult to understand. I recommend leaving only the WP names and providing the coordinates in the text or in the attachment.

- In Figures 8, 10, 12, and 14 it is not clear what the white dotted line means. I recommend supplementing the pictures with a legend.

 - The quality of the pictures needs to be improved.

- Tables 1, 2, 3, and 4 Values are presented with 6-digit precision. What is the point of such precision if all 5 digits are equal to 0?

- Subsection 4.5 does not relate to employment.

Abstract conclusions are presented. There is no clear validation of the obtained results with similar works of other authors.

Author Response

Thank you for your suggestion. We have made revisions to our manuscript based on your suggestions, details are given in the appendix.

Reviewer 2 Report

The paper “Acquisition and processing of UAV fault data based on Time Line Modeling method” proposes a Time line method to acquire and process the UAV fault data.

1.       Abstract is weakly written with stray information. Please clearly mention the motivation of your proposed work clearly.

2.       References must be cited properly in the text. In this regard, please follow the journal guidelines.

3.       Figure 1 is blurred.

4.       Section 3 is about the Proposed methodology. The methodology of proposed method of acquisition and processing must also be explained in a flow diagram.

5.       The captions of figures and tables are not elaborative like Figure 2 refers to the images before and after interpolation. The same should be explained in the caption to make it easy for the reader to understand. The same is for figure 5.

6.       What is the significance of figure 21. It is unclear from the caption what actually its showing.

7.       The overall structure of the paper can be improved by properly arranging the sections and subsections.

8.       Please explain your contribution in terms of classifying UAV fault data too.

Author Response

(The authors gave the same response as above.)

Round 2

Reviewer 1 Report

Thanks for the corrections.

A few inaccuracies remain:

- Figures 3, 17, 18, and 19 are blurred, hard to read, or illegible symbols. I recommend you go through all the figures and improve their quality so that all the information is legible in the printed version.

Author Response

Thank you very much for your suggestion, we have modified your question accordingly. We have written the details in the attachment.

Reviewer 2 Report

The paper is revised significantly with approximately all the changes made. However, few issues must be addressed before the acceptance of the manuscript.

1.       Please explain the figure 1. As what would be output of the detection architecture.

2.       Line 149: authors state that the data obtained after application of TLM method can effectively be applied on ML algorithms. It will be better if the same is shown in Flow diagram too.

3.       Section 3.1 can be named as Simulation environment and section 5 as experimentation.

4.       In Figures 3,5 and 6, subfigures must be captioned.

5.       Figure 18 : is it the percentage of each data label after processing? Please mention.

6.       Several figures are still not legible. Improve them.

Author Response

(The authors gave the same response as above.)
